# Effect of Neoadjuvant Chemotherapy on Tumor-Infiltrating Lymphocytes in Resectable Gastric Cancer: Analysis from a Western Academic Center

**DOI:** 10.3390/cancers16071428

**Published:** 2024-04-07

**Authors:** Elliott J. Yee, Danielle Gilbert, Jeffrey Kaplan, Sachin Wani, Sunnie S. Kim, Martin D. McCarter, Camille L. Stewart

**Affiliations:** 1Division of Surgical Oncology, Department of Surgery, University of Colorado, Anschutz Medical Campus, Aurora, CO 80045, USA; danielle.gilbert@cuanschutz.edu (D.G.); martin.mccarter@cuanschutz.edu (M.D.M.); camille.stewart@cuanschutz.edu (C.L.S.); 2Department of Pathology, University of Colorado, Aurora, CO 80045, USA; jeffrey.kaplan@cuanschutz.edu; 3Division of Gastroenterology, Department of Medicine, University of Colorado, Aurora, CO 80045, USA; sachin.wani@cuanschutz.edu; 4Division of Medical Oncology, Department of Medicine, University of Colorado, Aurora, CO 80045, USA; sunnie.kim@cuanschutz.edu

**Keywords:** gastric cancer, tumor-infiltrating lymphocytes, neoadjuvant chemotherapy, immunotherapy

## Abstract

**Simple Summary:**

In this investigation, we analyzed the number, type, and location of immune cells within surgically resected gastric cancer specimens treated with or without preoperative chemotherapy. We hypothesized that chemotherapy can stimulate the host immune system, as evidenced by an increased number of anti-tumor infiltrating lymphocytes in the tumor microenvironment. We found significantly elevated levels of immune cells within chemotherapy-treated tumors compared with chemotherapy-naïve specimens. We also revealed important associations between survival and immune lymphocytes in the tumor-related stromal tissue. Together, we added evidence supporting the immunostimulatory role of chemotherapy and underscore the potential utility of immunotherapy in resectable gastric cancer.

**Abstract:**

Tumor-infiltrating lymphocytes (TILs) are an emerging biomarker predictive of response to immunotherapy across a spectrum of solid organ malignancies. The characterization of TILs in gastric cancer (GC) treated with contemporary, multiagent neoadjuvant chemotherapy (NAC) is understudied. In this retrospective investigation, we analyzed the degree of infiltration, phenotype, and spatial distribution of TILs via immunohistochemistry within resected GC specimens treated with or without NAC at a Western center. We hypothesized that NAC executes immunostimulatory roles, as evidenced by an increased number of anti-tumor TILs in the tumor microenvironment. We found significantly elevated levels of conventional and memory CD8+ T cells, as well as total TILs (CD4+, CD8+, T_reg_, B cells), within chemotherapy-treated tumors compared with chemotherapy-naïve specimens. We also revealed important associations between survival and pathologic responses with enhanced TIL infiltration. Taken together, our findings advocate for an immunostimulatory role of chemotherapy and underscore the potential synergistic effect of combining chemotherapy with immunotherapy in resectable gastric cancer.

## 1. Introduction

Gastric cancer (GC) is among the most common and aggressive gastrointestinal (GI) cancers worldwide [1,2]. While accounting for only 1.5% of new cancer diagnoses in the United States, nearly half of patients present with advanced disease [3]. Although perioperative chemotherapy regimens have evolved and garnered modest improvements in OS when administered in the neoadjuvant setting, the 5-year survival in advanced GC remains less than 40% [4]. Thus, the need for improved anti-tumor therapies for gastric cancer is paramount.

Immunotherapy, specifically immune checkpoint blockade (ICB), has revolutionized the care of several solid organ malignancies, such as cutaneous melanoma, non-small cell lung cancer (NSCLC), and renal cell carcinoma [5,6,7,8]. Recent randomized control trial (RCT) data have established adjuvant ICB therapy in resected stage II/III esophageal cancer with residual disease in the surgical specimen as the standard of care in light of significantly prolonged disease-free survival with immunotherapy [9]. While the consensus guidelines currently recommend ICB immunotherapy in unresectable or metastatic GC that harbor established biomarkers predictive of a response to immunotherapy, its utility for potentially resectable GC tumors warrants further investigation [10].

It is well known that the degree and phenotype of tumor-infiltrating lymphocytes (TILs) is a prognostic marker for the response to ICB [11,12,13,14]. In triple-negative breast cancer and NSCLC, higher cytotoxic T cells (CD8+ T cells) demonstrate higher rates of overall response to ICBs, along with improved progression-free and overall survivals (PFS and OS, respectively) compared with those with lower CD8+ T cells [15,16]. It is also known that conventional chemotherapeutic agents, such as anthracyclines and platinum-based agents, the latter of which are frequently used to treat GC, can favorably alter the tumor microenvironment (TME) by inducing immunogenicity and synergizing the anti-tumor effect of host immunostimulatory agents [17,18].

The effect of contemporary multiagent chemotherapy on the degree of infiltration, phenotypes, and spatial distribution of TILs in potentially resectable GC is not well defined. The current body of work lacks an analysis of memory immune cell subtypes and the consideration of spatial (intratumoral versus stromal) TIL distributions [17,19,20]. Furthermore, most studies that do report on GC are from Asia, which is known to have a distinct disease biology treated with different chemotherapeutic regimens than in the West [21]. Considering these differences, we sought to characterize the density and infiltrative patterns of conventional and memory TIL subtypes of GC treated with or without chemotherapy at a Western academic referral center. We hypothesized that chemotherapy favorably alters the TME of GC, leading to increased levels of anti-tumor TILs.

## 2. Methods

### 2.1. Patient Cohort

After obtaining institutional review board consent, all adult patients with a biopsy-proven diagnosis of gastric adenocarcinoma who ultimately underwent a resection with curative intent from 2012–2020, either endoscopically or surgically, at our institution and had available formalin-fixed, paraffin-embedded (FFPE) tissue samples for histologic analysis were included in this study. A retrospective review of a prospectively maintained, clinically oriented database of patients was conducted. Review of the patient electronic health record was performed for missing data. After the patients were identified, additional FFPE slides were requested from areas of invasive tumor that were at least 2 mm in diameter. Slides were reviewed during creation, and the areas of invasive disease were determined by a board-certified gastrointestinal pathologist (author J.K.).

### 2.2. Definitions and Immunologic Profile Characterization

TIL populations were characterized by the multiplex immunohistochemistry (IHC) staining of associated cell surface (cluster of differentiation (CD)) or intranuclear markers using the Vectra-7-tumor-infiltrating lymphocyte kit (PerkinElmer, Waltham, MA, USA). The included TILs and markers were as follows: B cells/CD220+, CD8 T cells/CD8+, CD4 T cells/CD4+, T regulatory (T_reg_) cells/forkhead box P3 (FOXP3)+, CD8 memory T cells/CD8+/CD45RO+, CD4 memory T cells/CD4+/CD45RO+, memory Treg cells/CD4+/FOXP3+/CD45RO+, memory B cells/CD220+/CD45RO+, and epithelial malignant cell/pan cytokeratin. TIL density was defined as the number of above-stained immune cells per mm^2^ designated within the tumor or stroma regions of the tissue section. Total TILs were defined as the sum of the CD4+, CD8+, T_reg_, and B cells. Categorical assignment of high and low TIL densities was determined by the median value from the overall cohort. Clinical and pathologic staging of GC tumors were based on the latest National Comprehensive Cancer Network (NCCN) guidelines [10]. Pathologic assessment of the chemotherapy response score (CRS) was assessed per standard College of American Pathologists reporting conventions [22,23].

To characterize the immunologic profile of GC tumors, we also analyzed the EBV status, mismatch repair (MMR) protein expression, and tumor cell PD-L1. EBV status was determined by the in situ hybridization (iSH) detection of EBV-encoded small RNA (EBER)-positive tumor cells (ARUP Laboratories, Salt Lake City, UT, USA). Assessment of mismatch repair (MMR) protein expression was performed via immunohistochemistry (IHC) analysis of *MLH1*, *PSM2*, *MSH2*, and *MSH6* proteins (Leica; Wetzlar, Germany); deficiency (dMMR) was defined as a loss of >95% of any one of the protein expressions in tumor cells. Programmed Death-Ligand 1 (PD-L1, (clone E1L3N) Cell Signaling Technology; Danvers, MA, USA) expression was measured via the combined positivity score (CPS), which is defined as the number of positive PD-L1 stained cells via IHC divided by the total number of tumor cells multiplied by 100; values greater than 1 were considered representative of positive expression (Leica; Wetzlar, Germany).

### 2.3. Multiplex Immunohistochemistry

IHC was performed using an autostainer and then slides were reviewed using image-processing software by following a previously employed protocol [22]: Vectra 3.0 Automated Quantitative Pathology Imaging System (PerkinElmer) was used with the Bond RX autostainer (Leica). Slides were deparaffinized, heat treated in epitope retrieval solution 2 (ER2) antigen retrieval buffer for 20 min at 93 °C (Leica); blocked in antibody (Ab) Diluent (PerkinElmer); and incubated for 30 min with the primary antibody, 10 min with horseradish peroxidase-conjugated secondary polymer (anti-mouse/anti-rabbit, Perkin Elmer), and 10 min with horseradish peroxidase-reactive OPAL fluorescent reagents (Perkin Elmer). Slides were washed between staining steps with Bond Wash (Leica) and stripped between each round of staining with heat treatment in an antigen retrieval buffer. After the final staining round, the slides were heat treated in an antigen retrieval buffer, stained with spectral 4′,6-diamidino-2-phenylindole (PerkinElmer), and cover slipped with Prolong Diamond mounting media (ThermoFisher; Waltham, MA, USA). Whole-slide scans were collected using the 10× objective at a resolution of 1.0 μm. Then, 10 regions of interest identified by a gastrointestinal subspecialty trained board-certified pathologist (author J.K.) were scanned for multispectral imaging with the 20× objective at a resolution of 0.5 μm. The multispectral images were analyzed with inForm software (version 2.3, PerkinElmer) to unmix adjacent fluorochromes; subtract autofluorescence; segment the tissue into tumor regions and stroma; segment the cells into nuclear, cytoplasmic, and membrane compartments; and phenotype the cells according to cell marker expression.

### 2.4. Statistical Analysis

Parametric and nonparametric data are presented as means with standard deviations and medians with interquartile range, respectively. Categorical variables are expressed as absolute and relative frequencies (count and number). Categorical variables were compared using the chi-squared test; for continuous variables, parametric data were analyzed via Student’s t-test and non-parametric data with the Mann–Whitney U test. Comparison of more than two groups of non-parametric data was performed via the Wilcoxon sign-ranked test. Kaplan–Meier survival curves were generated to estimate time-to-event analyses for OS and RFS. All statistical analyses were performed in IBM SPSS version 28.0 (IBM, Chicago, IL, USA). Figures were constructed with SPSS or GraphPad Prism (version 10.0.0 for Windows, GraphPad Software, Boston, MA, USA). Quantification of the IHC staining of the MMR, PD-L1, and TIL densities was completed with inform Imaging Analysis Software (Akoya Biosciences, Marlborough, MA, USA). Statistical significance was considered *p* ≤ 0.05.

## 3. Results

### 3.1. Patient Cohort

Demographic and clinicopathologic variables of the entire patient cohort, stratified by the receipt of NAC, are displayed in Table 1. In total, 80 patients were identified, 68 of which provided pathologic specimens suitable for histologic analysis. Most patients were male (59%, *n* = 40), of Caucasian race (67%, *n* = 46), with a mean age of 63 years at the time of diagnosis (range 28–87 yrs, SD +/−15 yrs). All tumors were adenocarcinoma in origin. In the total cohort, most patients harbored clinical stage T3 tumors (52%, *n* = 35) and node-negative disease (N0 57%, *n* = 39). Nearly 75% of patients received NAC (*n* = 50), with the most common regimen being combination folinic acid, fluorouracil, and oxaliplatin (FOLFOX, 38%, *n* = 26). Neoadjuvant radiation was given to four percent of patients (*n* = 3). Surgical resection consisted of total gastrectomy or subtotal gastrectomy in 93% of patients (*n* = 63), while the remaining 7% underwent endoscopic resection (*n* = 5). Half of the cohort received adjuvant chemotherapy (52%, *n* = 35).

### 3.2. Demographic and Clinicopathologic Characteristics of Upfront Surgery and NAC Cohorts

Patients who received NAC were significantly more likely to have clinically larger tumors and node positive disease resulting in higher overall clinical stage (Table 1). Of the overall study cohort, 84% of patients (*n* = 57) met the current NCCN recommendations to receive preoperative chemotherapy (≥T2N0-3); of these patients, 19% did not receive NAT (*n* = 11), which was most commonly due to patient preference (55%, *n* = 6) in the setting of cT2N0 disease. In the ≥cT2N0-3 cohort, those who received NAC were more likely to have a positive node disease, proximal tumor location, and poor histologic grade.

### 3.3. TIL and Molecular Profiles of Upfront Surgery and NAC Cohorts

The intratumoral and stromal TIL phenotypes/densities and molecular profiles of the study cohort are detailed in Table 2 and Figure 1. In the overall cohort, patients who received NAC had significantly higher intratumoral conventional CD8+ T cells (14.3 vs. 5.1, *p* = 0.024) and total TILs (summation of CD4+, CD8+, T_reg_, B cells; 19.3 vs. 7.9; *p* = 0.047). There were no significant differences in the TIL densities in the tumor stroma between the two groups. Of note, there were no statistical differences in the intratumoral or stromal TIL densities in tumors treated with anthracycline- compared with platinum-based NAC regimens. The prevalences of EBV-positive, dMMR, and PD-L1-positive status were not different between the upfront surgery and NAC groups.

In the subset of patients with ≥cT2N0-3 disease, conventional CD8+ T cells (14.2 vs. 3.6) and total conventional TILs (18.8 vs. 6.7, *p* = 0.041) continued to be significantly upregulated in the tumor tissue of those who underwent NAC. Additionally, in this select cohort, intratumoral memory CD8+ T cells (2.0 vs. 0.7, *p* = 0.050) and total memory TILs (0.2 vs. 0.05, *p* = 0.048) were increased in tumors treated with NAC. Again, no difference in the TIL densities in the stromal component nor molecular phenotypes (EBV, MMR, and PD-L1 positivity) was found between the two cohorts. Although the CD8+ T cell to T_reg_ ratio was substantially increased in the tumor tissue of patients who received NAC, the difference only trended toward statistical significance (25.5 vs. 7.5, *p* = 0.079).

### 3.4. TIL Density and Oncologic Outcomes

The median follow-up time in the overall cohort was 43 months (range 30–65 mos) with death occurring in nearly half the overall cohort (47.1%, *n* = 32) and distant recurrence in over a third of patients (36.8%, *n* = 25). Peritoneal dissemination was the most common form of metastasis (11/25, *n* = 11). In both the overall and ≥cT2N0-3 cohorts, various high (defined as upper half from median value) TIL populations in the stromal but not intratumorally were associated with a significantly longer OS and RFS in the log-rank analysis. Figure 2 and Appendix A display the statistically significant Kaplan–Meier curves stratified by TIL phenotype with the associated log-rank analyses estimating the median for the OS and RFS. Although these specific high TIL densities were significantly associated with oncologic outcomes in the univariable analysis, significance was lost in the multivariable Cox regression models for the OS and RFS (Appendix A).

### 3.5. TIL Density and Pathologic Response

Most patients who underwent NAC demonstrated a poor pathologic response to preoperative treatment (chemotherapy response score of 1) (Table 1). There were no differences in response by chemotherapy regimen. No significant associations were observed between the high/low TIL categories and the pathologic response based on the median cutoff values; however, we found that the top quartile of densities of the intratumoral CD8+ T cells (OR 4.976; CI 1.166–21.242; *p* = 0.030) and total TILs (OR 6.667; CI 1.269–35.035; *p* = 0.025) were associated with were significantly associated with higher rates of near-complete and moderate responses (chemotherapy response scores of 3 and 2, respectively) compared with a poor response. Similarly, stromal CD8+ conventional T cells (OR 11.812; CI 1.3254–103.038; *p* = 0.025), CD8+ memory T cells (OR 14.0; CI 1.615–121.369; *p* = 0.017), total TILs (OR 5.625; CI 1.062–29.799; *p* = 0.042), and total memory TILs (OR 14.0; CI 1.615–121.369; *p* = 0.017) were more likely to be associated with an improved pathologic response.

## 4. Discussion

In the present investigation, we compared TIL phenotypes and infiltrative patterns in resected GC specimens from patients who did and did not undergo NAC. We hypothesized that among our cohort of patients treated at a Western academic center, NAC-treated tumors would demonstrate higher TIL densities in the TME compared with non-NAC counterparts. We found that in both the overall cohort and among those recommended to receive NAC (≥cT2N0-3), the tumors from NAC recipients demonstrated significantly increased intratumoral, but not stromal, TILs compared with patients that forewent NAC. Furthermore, we observed an improved OS, RFS, and pathologic response in patients with high compared with low TIL infiltration who received NAC.

Based on the results of recent RCTs, the application of immunotherapy in GC has been limited to unresectable or metastatic disease harboring specific immunotherapy-responsive molecular phenotypes, e.g., PD-L1 positive, MSI-H, and TMB-H [7,24,25]. The results of such trials have raised the potential that ICB therapy could be beneficial for resectable GC. The only published report from a phase III RCT that utilized combined chemotherapy plus ICB versus chemotherapy plus placebo for locally advanced GC/gastroesophageal junction (GEJ) tumors did not show a statistical difference in event-free survival at a median follow-up of nearly fifty months but did demonstrate a significant improvement in pathologic complete response with combination chemotherapy and ICB [26]. Recently, the phase III CheckMate-577 trial in resected esophageal/GEJ tumors reported a significantly longer disease-free survival in patients treated with adjuvant nivolumab compared with a placebo [9]. Notably, these improved outcomes occurred independently of the PD-L1 status, which is a finding that highlights alternative prognostic biomarkers predictive of response to immunotherapy. One such biomarker may be the degree of anti-tumor TIL infiltration within the TME [13,27]. Higher intratumoral and stromal TIL infiltrate, particularly cytotoxic CD8+ T cells, have been associated with longer survival and higher rates of pathologic response after ICB therapy compared with those with lower TIL infiltrate in advanced solid organ tumors [11,13,16,28]. Therefore, identifying mechanisms to increase tumor-targeting TIL populations into the TME may facilitate immunotherapy in resectable GC.

The use of NAC has become the standard of care for localized GC [29,30]. Mounting evidence suggests that while conventional chemotherapeutic agents play various immunosuppressive roles, they may also induce substantial immunogenicity and immunostimulation against malignancy by producing tumor-derived neoantigens, improving cytotoxic T cell recognition of tumor cells, and upregulating the damage associated molecular patterns (DAMPs) and cell surface molecules recruiting effector cells to the TME [17,31]. However, the data demonstrating the impact of contemporary, multiagent chemotherapy on the degree and phenotypes of TILs in GC are lacking. Thus, we aimed to analyze the TIL composition in GC tumors treated with and without NAC.

We found that postoperative GC specimens treated with NAC demonstrated significantly increased densities of intratumoral TILs compared with those that did not undergo NAC. In the overall cohort, which included patients with overall clinical stage I-III disease, CD8+ conventional T cells and total TILs were substantially elevated in NAC-exposed tumors. For those whom NAC was recommended per the NCCN guidelines (clinical stage ≥T2N0-3), the upregulation of TILs was even more widespread, as both conventional and memory subtypes of CD8+ T cells and total TILs were increased within the tumor tissue. Notably, we did not find differences in the stromal TIL densities between the two groups, although both conventional CD4+ and CD8+ T cells were at least twofold greater in the NAC cohort. Our observations that anti-tumor TILs were increased after NAC is consistent with the present literature in a range of epithelial carcinomas, including breast, non-small cell lung cancer, colorectal, and ovarian [32,33,34,35,36]. Our findings also corroborate those of Yu et al., who reported increased CD4+ and CD8+ T cell populations in Asian patients after receiving a combination of preoperative 5-FU and platinum-based agent, with or without a taxane and gastrectomy [19]. Unlike Xing et al. and Hu et al., we did not find a significant difference in the intratumoral or stromal T_reg_ cells, which may be secondary to known differences in Western versus Asian gastric cancer biology and differences in NAC regimens [20,37].

Notably, to our knowledge, we are the first to report the relationship between increased memory T cell infiltration and the receipt of NAC in GC. Memory subtypes are known to play important roles in executing a durable anti-tumor response [38]. Furthermore, recent preclinical data suggests that neoantigen stimulation of CD4+ T cells can facilitate the generation of specialized memory CD4+ T cells that can be utilized in adoptive T cell immunotherapy to prime effector CD8+ T cells in mitigating metastasis [39]. Lastly, both clinical and preclinical studies showed that the response to ICB is positively related to the proportion of memory T cells, suggesting the importance of memory phenotypes to mediating host immune response [40,41]. Taken together, we show that memory T cell subtypes were higher in the NAC-treated tumors, which may portend improved tumor control with IT.

In addition to the enhanced TIL infiltration in the NAC-treated specimens, we identified associations between high TIL phenotypes and an improved OS and RFS in patients who received NAC. Interestingly, despite observing statistically significant higher densities of intratumoral TILs between the NAC and upfront surgery cohorts, survival associations were only related to high stromal rather than intratumoral TILs. These findings support existing literature citing similar associations with higher stromal TILs and an improved RFS with breast and ovarian carcinomas [11,19,33,42]. Further, stromal TILs, particularly CD8+ T cells, were proposed to be a stronger prognostic biomarker of the response to ICBs and survival than intratumoral TILs, as reported by a meta-analysis including 2559 patients with a variety of solid organ tumors treated with immune checkpoint inhibitors [11]. Potential explanations for this finding may be that the intratumoral TILs, while increased, may be over-exposed to the tumor, rendering them an inactive, “exhausted” phenotype [43]. Additionally, active cytotoxic cells at the tumor periphery or invasive margin may be more proximal to antagonizing the aggressive metabolic and immune re-programming occurring at the tumor borders, thus critical to controlling tumor growth and dissemination [44]. To this point, higher stromal TILs in the primary tumor site were shown to correlate with a decreased metastatic burden, which is consistent with our associations between an improved RFS with increased stromal TIL populations [38,45]. Paradoxically, a high density of T_reg_ cells in the tumor stroma was also associated with improved recurrence-free survival in patients with ≥cT2N0-3 disease treated with NAC. This may have been a consequence of a generalized influx of immune cells after NAC and/or the body’s attempt to attenuate a heightened pro-inflammatory, anti-tumor response. Notably, we also observed an improved pathologic response to preoperative chemotherapy in a select subset of patients with the highest quartile of intratumoral- and stromal-infiltrating immune cells, supporting previous work demonstrating similar results in other NAC-treated carcinomas [46,47,48]. Nevertheless, given that distant metastases are the primary mode of failure for gastric cancer, there are evidently a multitude of mechanisms driving tumor immune evasion and progression that may be independent of the TILs that are associated with the primary tumor [49].

While this study added a novel perspective to the immunity landscape of resectable GC after NAC in Western patients, our results should be considered in the context of its limitations. As a retrospective, single-center endeavor, this study was constrained by inherent selection bias, a small sample size, and heterogeneity in administered chemotherapy regimens and data collection/reporting. We recognize that the inclusion of seven patients with metastatic disease per pathologic staging may add to the heterogeneity of our results; however, all metastatic patients had either intraoperative hepatic or peritoneal frozen biopsies that demonstrated no overt evidence of malignancy, and thus, they would have clinically achieved resection given the available evidence at the time of surgery. Additionally, we recognize our TIL and immunologic profile characterization is far from exhaustive, yet we aimed to bridge gaps according to prior literature. In our spatial TIL analysis, while we added novelty in differentiating intratumoral and stromal TILs, we did not assess TILs specifically confined to the tumor invasive margin, which is a metric that has risen to certain prognostic value. Due to the retrospective, clinically oriented nature of this study, we were not able to fully explain the relationships between the intratumoral and stromal TILs with long-term oncologic outcomes. Lastly, while a strength of this study was the in-depth nature of our analysis of TILs in the TME of Western GC, our results may not be fully translatable to GC at large considering that the GC that arises in Asia is known to be biologically distinct. Given these limitations, future work should be dedicated to prospective, protocol-based analysis that further details specific TILs, such as granzyme B CD8+ T cells, effector and central memory T cells, natural killer cells, and those of the “exhausted” phenotype.

## 5. Conclusions

The immunity TME of GC is highly heterogenous. Identifying mechanisms to facilitate novel therapeutics, i.e., immunotherapy, in an effort to improve the outcomes in GC is paramount. In this investigation, we observed that resected GC treated with NAC boasted higher intratumoral TILs, namely, conventional CD8+ and total TILs, compared with tumors that underwent upfront surgery across all clinical stages of localized disease. Importantly, we also established that the memory subtypes were upregulated in a subset of higher-stage patients who met the consensus criteria for NAC. Furthermore, we highlighted the prognostic value of stromal rather than intratumoral TILs for patients with GC undergoing NAC. Together, our novel findings affirm the need for further investigation into the complex interplay between the TME, TILs, chemo-, and immunotherapy.

## Figures and Tables

**Figure 1 cancers-16-01428-f001:**
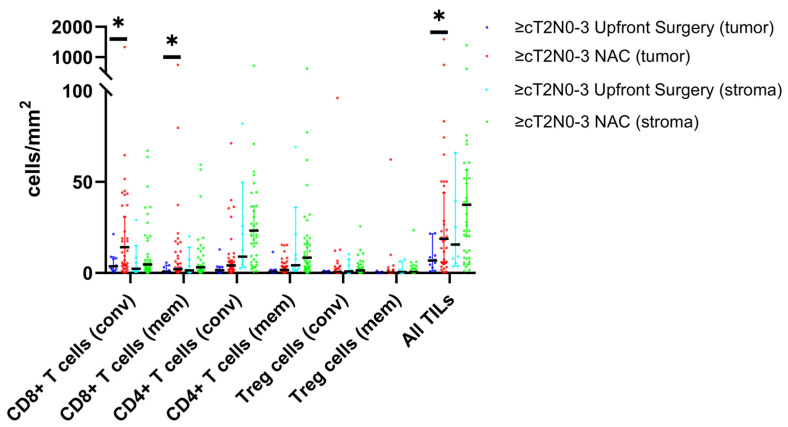
Scatter plot of tumor-infiltrating lymphocyte (TIL) densities in ≥cT2N0-3 disease stratified by receipt of neoadjuvant chemotherapy (NAC) and tissue location (intratumoral vs. stromal). *Conv*, conventional; *mem*, memory; *Treg*, T regulatory. Black line represents median with error bars for 95% confidence interval. * Statistical significance *p* < 0.05.

**Figure 2 cancers-16-01428-f002:**
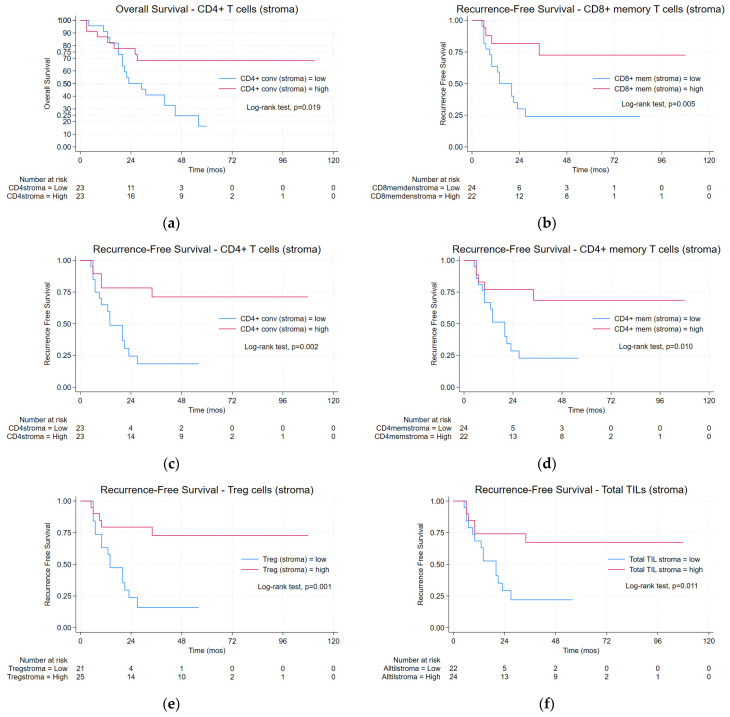
(**a**–**f**) Kaplan–Meier survival curves with log-rank tests demonstrating longer (**a**) overall survival and (**b**–**f**) recurrence-free survival with high vs. low stromal TILs in patients with ≥cT2N0-3 disease treated with NAC. Median survival follows in parentheses: (**a**) low (29.0 mos) versus high (not reached (NR)) CD4+ T cells, *p* = 0.019; (**b**) low (14.0 mos) versus high (NR) CD8+ memory T cells, *p* = 0.005; (**c**) low (14.0 mos) versus high (NR) CD4+ T cells, *p* = 0.002; (**d**) low (20.0 mos) versus high (NR) CD4+ memory T cells, *p* = 0.010; (**e**) low (14.0 mos) versus high (NR) T regulatory cells, *p* = 0.001; (**f**) low (20.0 mos) versus high (NR) total TILs, *p* = 0.011.

**Table 1 cancers-16-01428-t001:** Demographic and clinicopathologic characteristics of the overall cohort and of patients with ≥cT2N0-3 disease stratified by the receipt of neoadjuvant chemotherapy (NAC). EMR: endoscopic mucosal dissection; ESD: endoscopic submucosal dissection; CRS: chemotherapy response score; US: upfront surgery; FOLFOX: 5-FU, leucovorin, and oxaliplatin; FLOT: 5-FU, leucovorin, oxaliplatin, and docetaxel; EOX: epirubicin, oxaliplatin, and capecitabine; DCF: docetaxel, cisplatin, and 5-FU; ECF: epirubicin, and cisplatin, 5-FU; ECX: epirubicin, cisplatin, and capecitabine.

Characteristic	Overall Cohort (*n* = 68)	Upfront Surgery (*n* = 18)	NAC (*n* = 50)	*p*-Value	≥cT2N0-3 US (*n* = 11)	≥cT2N0-3 NAC (*n* = 46)	*p*-Value
**Demographic Characteristics**							
Sex, *n* (%) Male Female	40 (58.8) 28 (41.2)	8 (44.4) 10 (55.6)	32 (64.0) 18 (36.0)	0.148	7 (63.6) 4 (36.4)	30 (65.2) 16 (34.8)	0.921
Age at diagnosis, mean (SD)	62.8 (53.3, 73.3)	64.6 (+/−18.0)	65.5 (+/−13.5)	0.531	69.8 (+/−16.6)	63.0 (+/−13.4)	0.156
Race, *n* (%) White Black/African American Asian American Indian/Alaskan native Other	46 (67.6) 6 (8.8) 8 (11.8) 1 (1.5) 7 (10.3)	13 (72.2) 1 (5.6) 2 (11.1) - 2 (11.1)	33 (66.0) 5 (10.0) 6 (12.0) 1 (2.0) 5 (10.0)	0.944	7 (63.6) 1 (9.1) 1 (9.1) - 2 (18.2)	31 (67.4) 4 (8.7) 6 (13.0) - 5 (10.9)	0.944
ECOG status 0 1 2	36 (52.9) 28 (41.2) 4 (5.9)	7 (38.9) 9 (50.0) 2 (11.1)	29 (58.0) 19 (38.0) 2 (4.0)	0.285	4 (36.4) 5 (45.5) 2 (18.2)	26 (56.5) 18 (39.1) 2 (4.3)	0.202
**Clinicopathologic Characteristics**							
Clinical T stage, *n* (%) T1a T1b T2 T3 T4 T4a T4b Missing	3 (4.4) 7 (10.3) 12 (17.6) 35 (51.5) 1 (1.5) 7 (10.3) 2 (2.9) 1 (1.5)	2 (11.1) 5 (27.8) 7 (38.9) 3 (16.7) - - 1 (5.6) -	1 (2.0) 2 (4.0) 5 (10.0) 32 (64.0) 1 (2.0) 7 (14.0) 1 (2.0) 1 (2.0)	**<0.001**	- - 7 (63.6) 3 (75.0) - - 1 (9.1) -	- - 5 (10.9) 32 (69.6) 1 (2.2) 7 (15.2) 1 (2.2) -	**<0.001**
Clinical N stage, *n* (%) N0 N1-2 Missing	39 (57.4) 28 (41.2) 1 (1.5)	16 (88.9) 2 (11.1) -	23 (46.9) 26 (52.0) 1 (2.1)	**0.002**	9 (81.8) 2 (18.2) -	20 (43.5) 25 (54.3) 1 (2.2)	**0.026**
Overall clinical stage, *n* (%) Stage I Stage II Stage III Stage IVa Missing	19 (27.9) 21 (30.9) 25 (36.8) 2 (2.9) 1 (1.5)	14 (77.8) 2 (11.1) 1 (5.6) 1 (5.6)	5 (10.2) 19 (38.0) 24 (48.0) 1 (2.0) 1 (2.0)	**<0.001**	7 (63.6) 2 (18.2) 1 (9.1) 1 (9.1) -	3 (6.5) 18 (39.1) 24 (52.2) 1 (2.2) -	**<0.001**
Tumor location, *n* (%) Distal Proximal Linitis plastica Undefined	45 (66.2) 18 (26.5) 4 (5.9) 1 (1.5)	15 (75.0) 2 (10.0) - 1 (5.0)	30 (60.0) 16 (32.0) 4 (8.0) -	0.091	10 (90.9) - - 1 (9.1)	26 (56.5) 16 (34.8) 4 (8.7) -	**0.034**
Histologic subtype, *n* (%) Intestinal Diffuse/signet-ring Mixed Neuroendocrine	24 (35.3) 39 (57.4) 3 (4.4) 2 (2.9)	9 (50.0) 8 (44.4) - 1 (5.6)	15 (30.0) 31 (62.0) 3 (6.0) 1 (2.0)	0.277	7 (63.6) 4 (36.4) - -	15 (32.6) 28 (60.9) 2 (4.3) 1 (2.2)	0.275
**Preoperative and Intraoperative Characteristics**							
Neoadjuvant regimen, *n* (%) FOLFOX FLOT EOX DCF ECF ECX Other doublet combination (e.g., XELOX, CAPOX)	25 (36.8) 6 (8.8) 6 (8.8) 5 (7.4) 2 (2.9) 1 (1.5) 4 (5.9)	- - - - - - -	25 (36.8) 6 (8.8) 6 (8.8) 5 (7.4) 2 (2.9) 1 (1.5) 4 (5.9)	-	- - - - - - -	23 (50.0) 6 (13.0) 6 (13.0) 5 (10.9) 1 (2.2) 1 (2.2) 4 (8.7)	-
Rounds of chemotherapy	3.89 (+/−1.7)	-	-	-		4.0 (3.0–4.0)	-
Neoadjuvant radiation, *n* (%) No Yes	65 (95.6) 3 (4.4)	- -	45 (93.8) 3 (6.3)	0.288	11 (100)	43 (93.5) 3 (6.5)	0.288
Type of resection, *n* (%) Partial gastrectomy Total gastrectomy EMR/ESD	40 (58.8) 23 (33.8) 5 (7.3)	9 (50.0) 5 (27.8) 4 (20.0)	31 (62.0) 18 (36.0) 1 (2.0)	**0.019**	7 (63.6) 3 (27.3) 1 (9.1)	28 (60.9) 17 (37.0) 1 (2.2)	0.482
**Pathologic Tumor Characteristics**							
Pathologic overall stage, *n* (%) Stage I Stage II Stage III Stage IV	21 (30.9) 20 (29.4) 20 (29.4) 7 (10.3)	11 (61.1) 1 (5.6) 5 (27.8) 1 (5.6)	10 (20.0) 19 (38.0) 15 (30.0) 6 (12.0)	**0.006**	4 (36.4) 1 (9.1) 5 (45.5) 1 (9.1)	9 (19.6) 18 (39.1) 15 (32.6) 4 (8.7)	0.270
Clinical to pathologic stage change, *n* (%) No change Downstage Upstage Missing	35 (51.5) 17 (25.0) 15 (22.1) 1 (1.5)	13 (72.2) 1 (5.6) 4 (22.2) -	22 (44.0) 16 (32.0) 11 (22.0) 1 (2.0)	**0.058**	6 (54.5) 1 (9.1) 4 (36.4) -	21 (45.7) 16 (34.8) 9 (19.6) -	0.201
Histologic subtype, *n* (%) Intestinal Diffuse/signet-ring Mixed Neuroendocrine	24 (35.3) 39 (57.4) 3 (4.4) 2 (2.9)	9 (45.0) 10 (50.0) 1 (5.0)	15 (31.3) 29 (60.4) 3 (6.3) 1 (2.1)	0.443	7 (63.6) 4 (36.4) - -	15 (32.6) 28 (60.9) 2 (4.3) 1 (2.2)	0.275
Histologic differentiation, *n* (%) Poor Poor–moderate Moderate Mod to well Well	42 (61.8) 5 (7.4) 17 (25.0) 1 (1.5) 3 (4.4)	7 (38.9) 2 (11.1) 6 (33.3) 1 (5.6) 2 (11.1)	35 (70.0) 3 (6.0) 11 (22.0) - 1 (2.0)	0.075	3 (27.3) 2 (18.2) 5 (45.5) 1 (9.1) -	31 (67.4) 3 (6.5) 11 (23.9) - 1 (2.2)	**0.047**
Margin status, *n* (%) R0 R1 R2	58 (85.3) 10 (14.7) -	18 (100) - -	40 (80.0) 10 (20.0) -	**0.040**	11 (100) -	37 (80.4) 9 (19.6) -	**0.040**
Treatment effect, *n* (%) Minimal residual disease (CRS 3) Moderate response (CRS 2) Poor response (CRS 1) Unknown	4 (8.0) 21 (42.0) 22 (44.0) 3 (6.0)	-	4 (8.0) 21 (42.0) 22 (44.0) 3 (6.0)	-	-	4 (8.7) 20 (43.5) 19 (41.3) 3 (6.5)	-

**Table 2 cancers-16-01428-t002:** Molecular phenotype and tumor-infiltrating lymphocyte densities in the overall cohort (left) and in patients with ≥cT2N0-3 disease (right) stratified by location (intratumoral and stromal) and receipt of NAC. US, upfront surgery.

Molecular Phenotype and Tumor-Infiltrating Lymphocyte Profiles							
	Overall Cohort (*n* = 68)	Upfront Surgery (*n* = 18)	NAC (*n* = 50)	*p*-Value	≥cT2N0-3 US (*n* = 11)	≥cT2N0-3 NAC (*n* = 46)	*p*-Value
EBV status, *n* (%) Negative Positive	65 (95.6) 3 (4.4)	18 (100) -	47 (94.0) 3 (6.0)	0.288	11 (100) -	43 (93.5) 3 (6.5)	0.384
MMR, *n* (%) Proficient Deficient	60 (88.2) 8 (11.8)	17 (94.4) 1 (5.6)	43 (86.0) 7 (14.0)	0.340	10 (90.9) 1 (9.1)	40 (87.0) 6 (13.0)	0.720
PD-L1 status, *n* (%) Negative Positive	41 (60.3) 27 (39.7)	11 (61.1) 7 (38.9)	30 (60.0) 20 (40.0)	0.934	7 (63.6) 4 (36.4)	27 (58.7) 19 (41.3)	0.764
Tumor-Infiltrating Lymphocyte Densities—Intratumoral							
CD8+ T cells, cells/mm^2^ Conventional (CD8+) Memory (CD8+/CD45RO+)	8.6 (3.4, 37.1) 1.8 (0.8, 9.0)	5.1 (2.1, 8.5) 1.0 (0.6, 4.8)	14.25 (4.3, 43.7) 2.3 (1.1, 10.8)	**0.024** 0.119	3.6 (2.0, 8.1) 0.7 (0.4, 3.1)	14.2 (4.0, 43.7) 2.0 (1.0, 11.6)	**0.019** **0.050**
CD4+ T cells, cells/mm^2^ Conventional (CD4+) Memory (CD4+/CD45RO+)	3.4 (0.8, 8.0) 1.6 (0.4, 5.0)	1.7 (0.6, 5.7) 0.8 (0.3, 3.3)	4.5 (0.8, 9.2) 2.1 (0.6, 5.6)	0.182 0.254	1.5 (0.3, 3.2) 0.6 (0.2, 1.7)	4.2 (0.9, 8.3) 1.6 (0.5, 5.4)	0.089 0.119
T_reg_ cells, cells/mm^2^ Conventional (CD4+/FOXP3+) Memory (CD4+/CD45RO+)	0.4 (0.1, 1.7) 0.2 (0.04, 1.7)	0.5 (0.04, 1.5) 0.2 (0.02, 1.2)	0.4 (0.1, 1.9) 0.2 (0.04, 0.9)	0.671 0.950	0.3 (0.1, 1.1) 0.1 (0.04, 0.4)	0.4 (0.1, 1.9) 0.2 (0.03, 0.8)	0.442 0.754
B cells, cells/mm^2^ Conventional (CD220+) Memory (CD220+/CD45RO+)	0.02 (0.003, 0.15) 0	0.01 (0.01, 0.09) 0	0.04 (0.0, 0.16) 0	0.550	0.01 (0.0, 0.01) 0.0	0.03 (0.0, 0.16) 0.0	0.088
All TILs (CD8+, CD4+, B cell)	13.6 (5.5, 49.6)	7.9 (4.1, 15.4)	19.3 (5.6, 53.9)	**0.047**	6.7 (2.8, 9.6)	18.8 (5.4, 53.9)	**0.041**
ALL memory TILs	0.3 (0.03, 2.3)	0.11 (0.01, 0.49)	0.28 (0.05, 3.48)	0.098	0.05 (0.01, 0.3)	0.2 (0.04, 3.1)	**0.048**
CD8:Treg ratio	23.2 (6.6, 3.4)	7.8 (3.7, 53.3)	25.5 (12.4, 54.6)	0.123	7.5 (3.1, 29.5)	25.5 (7.0, 55.2)	0.079
Tumor-Infiltrating Lymphocyte Densities—Stromal							
CD8+ T cells, cells/mm^2^ Conventional (CD8+) Memory (CD8+/CD45RO+)	4.9 (1.6, 19.0) 3.3 (1.0, 11.6)	6.4 (1.2, 21.4) 4.4 (0.9, 13.8)	4.6 (1.8, 18.1) 3.1 (1.0, 8.8)	0.597 0.396	2.3 (1.1, 8.4) 1.45 (0.6, 7.3)	4.6 (1.9, 19.6) 3.1 (1.0, 10.2)	0.203 0.385
CD4+ T cells, cells/mm^2^ Conventional (CD4+) Memory (CD4+/CD45RO+)	23.2 (6.3, 53.1) 9.7 (2.4, 29.0)	23.6 (7.1, 73.7) 11.1 (3.0, 40.1)	23.2 (6.0, 50.4) 8.4 (2.3, 21.0)	0.906 0.359	8.9 (4.5, 21.2) 4.2 (1.8, 21.2)	23.2 (7.0, 54.2) 8.4 (2.6, 22.7)	0.143 0.454
T_reg_ cells, cells/mm^2^ Conventional (CD4+/FOXP3+) Memory (CD4+/CD45RO+)	1.3 (0.2, 3.4) 0.5 (0.1, 1.8)	1.0 (0.5, 4.9) 0.6 (0.2, 1.7)	1.4 (0.2, 2.8) 0.5 (0.1, 1.3)	0.592 0.254	0.8 (0.2, 4.1) 0.5 (0.1, 1.8)	1.4 (0.2, 3.3) 0.5 (0.1, 1.7)	0.716 0.952
B cells, cells/mm^2^ Conventional (CD220+) Memory (CD220+/CD45RO+)	1.2 (04, 6.7) 0.1 (0.02, 1.2)	1.3 (0.6, 10.0) 0.2 (0.02, 1.7)	0.9 (0.3, 5.6) 0.1 (0.2, 0.6)	0.294 0.555	0.9 (0.2, 1.3) 0.3 (0.02, 0.3)	1.1 (0.3, 6.6) 0.1 (0.02, 0.9)	0.379 0.201
All TILs (CD8+, CD4+, B cell)	36.1 (8.6, 74.9)	31.0 (8.7, 97.8)	37.4 (8.3, 71.1)	0.889	15.6 (5.3, 39.4)	37.4 (10.6, 73.3)	0.110
CD8:Treg ratio	3.6 (2.3, 10.3)	3.4 (1.4, 16.2)	3.6 (2.4, 9.6)	0.479	2.3 (1.2, 4.8)	3.6 (2.3, 9.6)	0.152

## Data Availability

The original contributions presented in the study are included in the article and Appendix A, further inquiries can be directed to the corresponding author.

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
