# Peer review of "Effect of Neoadjuvant Chemotherapy on Tumor-Infiltrating Lymphocytes in Resectable Gastric Cancer: Analysis from a Western Academic Center"

_cancers, 2024, doi:10.3390/cancers16071428_

Round 1

Reviewer 1 Report

Comments and Suggestions for Authors

I commend the authors on this well-written and timely study evaluating the role of tumor infiltrating lymphocytes after neoadjuvant chemotherapy in resectable gastric cancers.  I believe the conclusions of this study are well supported by the methodology and results, and the limitations of the study as a single-center retrospective study with relatively small sample size are clearly stated in the discussion.

Minor criticisms:

1. Table 1 clinical stage appears to be mislabeled for stage IVa, as it lists 67 patients with this stage.

2. Inclusion of patients with pathologic stage IV disease. This is typically considered unresectable disease, and the presence of metastatic disease could represent a different TIL profile.

3. It would be informative to see if there was a correlation between TIL and other factors, such as histologic subtype in diffuse/signet ring vs intestinal.

4. Formatting for the figures and tables should be reviewed, as some of the labelling and text is cut off. 

Author Response

Thank you very much for the comments/suggestions. Please see below for a point-by-point response.

  1. Thank you for identifying this error. We have amended the table to say "2 patients (2.9%)" with clinical stage IVa disease, which per NCCN guidelines reflects T4b disease. 
  2. Thank you so much for this comment. We agree with the reviewer that inclusion of patients with pathologic M1 disease may represent different tumor biology, however, we have elected to include these specific patients in the study cohort given these patients had no evidence of overt metastatic disease on preoperative imaging nor intraoperatively as the frozen biopsy samples returned without definitive diagnosis of carcinoma ie atypical cells thus we considered them clinically resectable at the time of surgery. We have added lines 370-374 to address this comment. 
  3. Thank you for this suggestion. We agree with the authors of the utility of comparing TIL populations between histologic subtypes. This is the focus of forthcoming work.
  4. Thank you for pointing this out. We have revised Figure 2 and Supplemental figures 1-2 KM curves for clarity and added number at risk tables. 

Reviewer 2 Report

Comments and Suggestions for Authors

Yee et al. retrospectively assessed the density, phenotype and spatial distribution as well as dynamics of TILs in a Western gastric cancer cohort (n=68) including patients that had been diagnosed/treated between 2012 and 2020 with either upfront chemotherapy (n=18) or “neoadjuvant” chemotherapy (n=50). Furthermore, the impact of TILs on clinical outcome (RFS, OS) and surrogate parameters (tumor regression grade) has been investigated. 

The authors reported a significant increase of itnratumoral TILS after neoadjuvant chemotherapy, mainly driven by conventional CD8+ T cell upregulation. The authors also investigated dynamics of biomarkers such as PD-L1 CPS and did not find changes during neoadjuvant chemotherapy. Dichotomous consideration (medians) of various TIL (subsets) in the stroma (but not in the tumor) were described to be prognostic (RFS, OS). The authors also report an association between intratumoral as well as stromal TILS and pathologic response when using other cut-offs (quartiles) for TIL categorization. 

Although the authors made a good effort with their investigation, the manuscript comes with several limitations which must be addressed prior to publication.

1.     A major limitation for any conclusion is the heterogeneity of the chemotherapy protocols 

2.     The majority of patients received FOLFOX, whereas FLOT represents the perioperative therapy of choice for most patients (which cannot solely by explained by the long inclusion period and “pre-FLOT”-era. Prior to the publication of the AIO-FLOT4 Trial, antracyclin-based triplet therapy was the standard perioperative regimen – why did so many patients only reiceive doublet chemotherapy? The median age argues against a “frail” population..

3.     (ECOG) performance score is not described in the baseline characteristics

4.     How many patient received post-operative chemotherapy (perioperative concept)

5.     Figure 2 – I suggest an improvement of the figures (mismatch of the color code, text within the graphs is hidden). Please show numbers at risk below the KM-curves, respectively. Please clarify/describe whether the baseline or post-neoadjuvant Chemotherapy TILs were used for this KM-curves

6.     Introduction: “Recent randomized control trial (RCT) 50 data has established adjuvant ICB therapy in resected stage II/III esophageal cancer as standard of care in light of significantly prolonged disease-free survival with immunotherapy”
 This is not for all-comers - please explain precisely the sub-cohort that receives adjuvant nivolumab in clinical practice

7.     Methods: “Clinical and pathologic staging of GC tumors were based on the latest National Comprehensive Cancer Network (NCCN) guidelines”
 Within this time frame (2012-2020) the staging classifciation has changed several times – please clarify whether the abovementioned sentence applies to all patients that had been included within this long time interval

8.     Please clarify with which clone the PD-L1 CPS was assessed. 

9.     Please cite the “chemotherapy response score” and explain why this score was used (it does not include pCR as a separate category?)

10.  Please provide in detail the chemotherapy protocols labeled “other” – this is important as one key point of the manuscript is the influence on TILs/PD-L1 density/expression

11.  How many patients received diagnostic laparoscopy as staging modality?

12.  How do the authors explain a favorable outcome with more Tregs in the stroma?

13.  I encourage the authors to analyze the prognostic impact of TILs (TIL subsets) in a multivariable analysis with other prognostic parameters (ECOG, stage, ..). Is it really an independent prognostic parameter?

Author Response

We thank the reviewer for their valuable contributions and suggestions to the paper. Please find a point-by-point response below.

  1. Thank you for this comment. We agree that the retrospective nature of this study spanning several eras of systemic chemotherapy recommendations for gastric cancer add heterogeneity to the NAC regimens administered which may have differential impact on TILs in resected specimens. We thus performed test of independence comparing TIL densities between anthracycline and platinum based NAC regimens and found no statistical differences between the two groups. We have added lines 184-186, page 6, and revised line 362 on page 11 to address this comment. 
  2. Thank you for this insightful comment. According to our colleagues in medical oncology, the practice at our administration shifted from triplet epirubicin based regimen to doublet therapy due to the toxicity-limiting effects of epirubicin (namely myeloid suppression/neutropenia) with evidence suggesting no difference in survival outcomes (https://pubmed.ncbi.nlm.nih.gov/28784312/, chrome-extension://efaidnbmnnnibpcajpcglclefindmkaj/https://www.e-crt.org/upload/pdf/crt-42-18.pdf, https://www.nature.com/articles/bjc2016126). We have added detailed NAC regimens in Table 1. 
  3. Thank you for this suggestion. We have added ECOG data to Table 1. 
  4. Thank you for this comment. Approximately half (51.5%) of patients received adjuvant chemotherapy, the most common regimen being FOLFOX (n=18, 27%). There were no significant associations between TILs high v low TIL densities between those who did and did not receive adjuvant chemotherapy. 
  5. Thank you for pointing this out. We have revised the KM curves for Figure 2 and Supplemental Tables 1-2 for clarity and inclusion of number at risk tables. We have added lines X in the manuscript to clarify that the TIL densities quantified were from post-neoadjuvant specimens; no pre-neoadjuvant samples were assessed for this study.
  6. Thank you very much for this comment. We have revised lines 51-52 on page 2 to say "...therapy in resected stage II/III esophageal cancer with residual disease in the surgical specimen as standard of care..." based on the inclusion criteria of the cited study. 
  7. Thank you for this comment. All clinical and pathologic staging has been updated to reflect the most current NCCN guidelines. 
  8. Thank you for this suggestion. We have added the PD-L1 clone number to lines 111-112 on page 3. 
  9. Thank you for this important comment. We have added lines 101-103, page 3 and associated citations regarding use of College of American Pathology gastric cancer reporting conventions, the most widely used reporting protocol used in the United States. Per these guidelines, pathologic complete response is classified as CRS0, none of which our patients were deemed to have.
  10. Thank you for this suggestion. We have revised Table 1 to include detailed display of neoadjuvant chemotherapy regimens. 
  11. Thank you for this important comment. 11 patients (12%) underwent staging laparoscopy. 
  12. Thank you very much for this interesting and pertinent suggestion. We posit that potentially receipt of neoadjuvant therapy broadly increases the immune infiltrate in and around the tumor, including anti-tumor CD8+ T cells as well as immunosuppressive Treg cells. Further, this could be the body's attempt to balance the increased pro-inflammatory, anti-tumor response via influx of CD8+ and FOXp3-/CD3+ T cells intratumorally and in the stroma. We have added lines 354-358 on page 11 to address this comment. 
  13. Thank you for this valuable suggestion. We agree that identifying granular factors influencing survival outcomes is important and meaningful to improving outcomes in gastric cancer. We have performed univariable and multivariable Cox regression survival analysis for both overall and recurrence free survival for the specific TILs that we found significant for improved survival on log-rank analysis. All measured TIL subtypes were significantly associated with improved survival on univariate Cox regression analysis but lost significance on multivariable analysis. We have added Supplemental Tables 1-2 displaying uni- and multivariable models. We have also added lines 214-216 to include the addition of these Cox models. Although not statistically significant in multivariable Cox regression analysis, we still advocate that high TIL densities may be potentially synergistic with other therapies such immunotherapy. 

Round 2

Reviewer 2 Report

Comments and Suggestions for Authors

The authors adequately addressed the raised points. However, prior to publication correct statistical methods have to be confirmed:

Point 5: The authors describe that the post-neoadjuvant TILS have been used for these KM-curves – please make sure that the TILS have been considered as a time-dependent variables.

Author Response

Thank you for this valuable comment. We did analyze the survival data in a time dependent fashion using Cox time dependent variable (TIL) models which resulted in the same statistical outcomes as Kaplan Meier log rank analysis. Time dependent analysis was more relevant for comparisons such as overall survival for CD4+ T cells (stroma) as the curves diverged later in the time course but still retained statistical significance using time dependent analysis.